Novel conditional tabular generative adversarial network based image augmentation for railway track fault detection

http://orcid.org/0000-0001-5429-9835 Raza Ali 1 ali.raza.scholarly@gmail.com
Sehar Rukhshanda 2
Moiz Abdul 2
http://orcid.org/0000-0001-6829-9705 Alluhaidan Ala Saleh 3 asalluhaidan@pnu.edu.sa
El-Rahman Sahar A. 4
http://orcid.org/0000-0003-0881-3164 AbdElminaam Diaa Salama 5 6
1 Department of Software Engineering, University of Lahore , Lahore , Pakistan
2 Institute of Computer Science, Khwaja Fareed University of Engineering and Information Technology , Rahim Yar Khan , Pakistan
3 Department of Information Systems, College of Computer and Information Sciences, Princess Nourah bint Abdulrahman University , Riyadh , Saudi Arabia
4 Computer Systems Program-Electrical Engineering Department, Faculty of Engineering-Shoubra, Benha University , Cairo , Egypt
5 MEU Research Unit, Middle East University , Amman , Jordan
6 Jadara Research Center, Jadara University , Irbid , Jordan
Yang Jiachen
Electronic publication date: 2025 Jun 18
Publication date: 2025
Volume: 11
Electronic Location ID: e2898
Received 2024 Oct 31; Accepted 2025 Apr 24
Copyright: © 2025 Raza et al.
Copyright year: 2025
Copyright holder: Raza et al.
License: This is an open access article distributed under the terms of the Creative Commons Attribution License, which permits unrestricted use, distribution, reproduction and adaptation in any medium and for any purpose provided that it is properly attributed. For attribution, the original author(s), title, publication source (PeerJ Computer Science) and either DOI or URL of the article must be cited.
License URL: https://creativecommons.org/licenses/by/4.0/

Keywords: Railway track fault, Fault detection, Machine learning, Deep learning, Generative AI, GAN

Funding: Princess Nourah bint Abdulrahman University, Riyadh, Saudi Arabia PNURSP2025R434 This research was funded by Princess Nourah bint Abdulrahman University Researchers Supporting Project number (PNURSP2025R234), Princess Nourah bint Abdulrahman University, Riyadh, Saudi Arabia. The funders had no role in study design, data collection and analysis, decision to publish, or preparation of the manuscript.

==============================
Railway track fault recognition is a critical aspect of railway maintenance, aiming to identify and rectify defects such as cracks, misalignments, and wear on tracks to ensure safe and efficient train operations. Classical methods for fault detection, including manual inspections and simple sensor-based systems, face significant challenges, such as high labour costs, human error, and limited detection accuracy under varying environmental conditions. These methods are often time-consuming and unable to provide real-time monitoring, leading to potential safety risks and operational inefficiencies. To address these challenges, efficient artificial intelligence-based image classification is being explored to enhance railway track fault detection accuracy, efficiency, and reliability. This research aims to develop an advanced generative neural network for efficient railway track fault detection. We propose a novel conditional tabular generative adversarial network (CTGAN)-based image augmentation approach to producing realistic synthetic image data using railway track images. We developed five advanced neural network techniques for comparison with railway track image classification. The random forest approach surpasses state-of-the-art studies with a high accuracy score of 0.99 for railway track fault detection. Hyperparameter optimization is applied to achieve optimal performance, and the performance is evaluated using the k-fold cross-validation approach. The proposed research enhances operational efficiency, reduces maintenance costs, and significantly improves the safety and reliability of rail transportation.

Introduction

Railway track faults encompass a variety of structural and material defects that occur within the railway infrastructure, which can compromise the safety and efficiency of train operations (De Bruin, Verbert & Babuvska, 2016). These faults typically manifest as physical anomalies on the track itself, including rail breaks, cracks, warping, and misalignments. Other common types of faults include issues with the ballast, such as contamination or uneven distribution, and problems with track geometry, such as incorrect spacing, alignment, or leveling of the rails (Lissandro, 2021). Faults may arise from natural wear and tear, environmental influences, improper maintenance, or material fatigue. Railway track failures are a significant cause of severe rail accidents, particularly in Pakistan, where train travel is prevalent. During the 2018–2019 period, an estimated 70 million passengers utilized this mode of transport, according to Pakistan Railways’ annual data (Qureshi, 2024).

Detecting and addressing railway faults promptly is crucial as they can lead to severe consequences, including derailments and other unsafe operating conditions (Tarawneh, Wilson & Porter, 2024). Detecting faults in railway tracks presents a host of challenges that stem from both the physical environment of the railways and the technical complexities involved in monitoring systems. Firstly, railway tracks often extend over vast and geographically diverse areas, which can include remote or inaccessible regions. This vastness not only complicates regular physical inspections but also requires extensive deployment of sensors and other monitoring technologies which can be costly and difficult to maintain (Alrahman & Adham, 2024). Traditional methods such as visual inspections and track circuit monitoring are often insufficient, prompting the need for advanced technologies like machine learning algorithms and predictive maintenance models.

Deep learning and generative adversarial network (GAN) based image processing (Yao, Jiang & Ng, 2024) have emerged as powerful tools in the detection of railway track faults, contributing significantly to the enhancement of railway safety and maintenance. Deep learning approaches, particularly convolutional neural networks (CNNs), excel in processing and analyzing vast amounts of image data, allowing for the precise identification of surface irregularities and structural deformities in railway tracks. These models automatically learn to detect subtle features such as cracks, misalignments, and wear that might be missed by traditional inspection methods. Furthermore, GAN, a variant of GANs tailored for tabular data, has been adapted for image-based applications to augment the dataset used for training deep learning models (He et al., 2024). GAN works by generating synthetic, yet realistic, images of faulty tracks under various conditions. This augmentation helps in creating robust models capable of generalizing well across diverse scenarios not covered by the limited available datasets. By integrating these technologies, the fault detection process becomes more automated and efficient, reducing the dependency on manual inspections and improving the predictive maintenance of railway infrastructure.

Our primary research contributions toward railway track fault detection are as follows: We proposed a novel CTGAN-based image augmentation approach to produce realistic synthetic image data using railway track images.

We developed five advanced neural network techniques in comparisons for railway track image classification.

Hyperparameters optimization is applied to find optimal performance and validate the performance using the k-fold validation approach.

The remaining section’s information on the research is structured as follows: “Literature Analysis” presents the literature analysis. “Materials and Methods” outlines the proposed methodology. “Results and Discussions” presents a comparative evaluation of the results obtained from the applied methods. The primary findings are summarized in “Conclusions”.

Literature analysis

This literature analysis focuses on railway track fault detection through machine learning (ML) techniques, provided in Table 1. A wide array of studies underscores the integration of advanced algorithms for enhancing predictive maintenance and safety measures. This body of research also delves into the comparative advantages of supervised learning approaches for preventing accidents and ensuring the smooth operation of rail services.

Table 1 The railway track fault detection related work summary analysis.

References	Year	Dataset used	Proposed technique	Performance score	Advantages & Disadvantages	
Chen et al. (2021)	2021	Self-Made	YOLOV3	87%	Low performance results, also the dataset was biased.	
Li et al. (2022)	2022	National academy of railway sciences test centre dataset	MBDA, YOLOV5S6, YOLOV5S, YOLOV5m, Faster RCNN R50, Faster RCNN R101	0.75 mAP, WBDA	Low mAP performance results during the detection.	
Nandhini & Mohammed Saif (2024)	2021	Kaggle	Weighted vector, Bayes SVM (LDA, PCA), CNN	89% accuracy Multi-scale CNN	Accuracy is good. However, classical methods were applied.	
Bhushan et al. (2017)	2017	Self-made	Sensors and GSM module	94.1% accuracy	Only sensors based dataset was utilized.	
Ritika & Rao (2018)	2018	Self-made	SunKink, inception3	97.5% precision SunKink	Good performance results, also the dataset was biased.	
Faghih-Roohi et al. (2016)	2016	Self-made	DCNN-small, DCNN-medium, DCNN-large	92% accuracy DCNN-Large	Moderate performance results, also the dataset was biased.	
Shafique et al. (2021)	2021	Self-made	SVM, DT, MLP, LR, RF, CNN	97% accuracy RF and DT		
He et al. (2021)	2022	Subway obstacle test dataset	RCNN	95%	More advanced neural networks can be built.	
Wang et al. (2024)	2024	Internal defects dataset	DL	93%	Only deep learning were applied.	
Li et al. (2024)	2023	Image data	Net-b7 model	85%	Accuracy performance results are very low.	

Chen et al. (2021) proposed a railway defects technique that is improved by the YOLOv3 algorithm, utilizing B-scan image recognition. In this experiment, 453 B-scan images were used. The proposed method automatically recognized box holes and accurately positioned a box in B-scan images. The study demonstrates that by employing the YOLOv3 algorithm, an accuracy of 87.41% was achieved.

Li et al. (2022) proposed a technique to enhance performance by utilizing multiple algorithms. They obtained features individually by employing various models such as neural networks (NN) to improve sub-networks combined in binary format. In this study, augmentation techniques were employed to enhance model accuracy. The data used in this study achieved an accuracy of 74%, which is higher than the mean average precision (mAP) of five compared to YOLOv3 algorithms.

Nandhini & Mohammed Saif (2024) proposed a CNN model for automatic railway track detection. In this experiment, vibration data is utilized for crack detection, with the dataset being sourced from Kaggle. Various machine learning approaches were employed in this study, resulting in an achieved accuracy of 89%.

Bhushan et al. (2017) proposed a system capable of detecting faults in railway tracks. In this experiment, an audio sensor and NS-AM type point machine were used to analyze obstacles, slackened nuts, and blasts. Two different experiments were conducted in this study to detect faults. The model achieved a 94% accuracy, demonstrating efficient results.

Ritika & Rao (2018) proposed a computer vision technique to detect railway faults. In this experimental study, images were captured at a rate of 30 frames per second using a camera. The ImageNet dataset was utilized for binary classification. The Inception v3 model was employed, specifically for classifying SunKink in the video. The model performed well on images, achieving an accuracy of 97%.

Faghih-Roohi et al. (2016) proposed the detection of railway faults using a dataset based on images. In this study, deep convolutional neural network (DCNN) was employed, specifically utilizing small, medium, and large DCNNs. Different sizes and activation functions characterized the various networks. Among the different DCNNs, the large deep convolutional neural network achieved an accuracy of 92%.

Shafique et al. (2021) aimed to enhance the detection of faults in old railway tracks by introducing an automatic recognition system. In this experiment, data from an old railway cart system was utilized. Three commonly found types of railway tracks were considered standard. Different machine learning algorithms were employed in this study, and an accuracy of 97% was achieved using random forest (RF) and DT.

He et al. (2021) proposed a technique to detect obstacles in railway tracks using the deep learning algorithm R-CNN. Previous techniques faced several issues, and this proposed technique aims to overcome those challenges. Features were extracted using a DCNN backbone in the experiments. The results of the research show that the deep learning algorithm RESNET 101 achieved an accuracy of 95%. The technical framework of detection is improved by enhancing detection speed.

Wang et al. (2024) proposed a technique for the recognition of internal rail defects using B-scan data images and deep learning algorithms. In this study, a railway fault detector was utilized to gather images of internal defects, incorporating both B-scan and actual defect images. The objective of the research is to create an automated system for detecting railway track faults. The results show that multiple deep learning algorithms are employed, achieving efficient outcomes.

Li et al. (2024) proposed to detect internal railway defects. In this study, a method based on CNN and customized image recognition is employed. In this experiment, features from image data are extracted. Our proposed model shows an accuracy of 85%, with evaluation metrics measuring precision at 81.71% and recall at 85%. Through the use of augmentation and transfer learning approaches, this model performs better and requires less data.

Existing work gaps

Previous studies on railway track fault detection primarily relied on classical data augmentation techniques. However, these methods often fail to capture complex variations in track faults, leading to lower detection performance. The limitations of traditional augmentation approaches highlight the need for more advanced techniques to enhance model robustness and improve fault detection accuracy.

Materials and methods

The materials and learning methods developed during this research are described in detail in this section. Additionally, the analysis of the dataset used to build learning models and the performance measures are thoroughly examined.

Our novel proposed methodology is depicted in Fig. 1. Initially, we acquired an image dataset of defective and non-defective railway tracks for research experiments. The track images are passed through a preprocessing stage to enhance prediction performance. In the next step, we applied a CTGAN-based image augmentation approach to improve the model’s generalizability. The augmented railway track data is partitioned into training and testing portions. We applied several neural network methods and evaluated them on the testing portion. The hyperparameterized AI model is then utilized for the early prediction of defective or non-defective railway tracks using track images.

Figure 1 The proposed railway track fault detection methodology.

Railway tracks images dataset

This research utilized a standard dataset (Adnan & Salman Ibne Eunus, 2024) based on railway track images. The track images in the dataset were resized to a shape of (224 × 224 × 3). The dataset contains both defective and non-defective railway track images. The sample track dataset is illustrated in Fig. 2. The dataset set contains a total of 384 railway tracks images. This dataset is utilized for the classification of defective and non-defective railway tracks using a vision-based neural approach.

Figure 2 The defective and non-defective railway tracks image analysis.

Railway tracks images preprocessing

We have applied several image-processing steps to the track images dataset. Initially, we removed images that were corrupted. The dataset folder contains two subfolders Defective and non-defective. We set the same size of the input images to a height of 256 pixels and a width of 256 pixels. Then, we looped over the label folders in the dataset folder. We appended each image and its label to the data and label lists. Finally, the pixels we read are converted into NumPy arrays for the data and label lists. The preprocessed dataset is then utilized to conduct our subsequent research experiments.

Novel proposed CTGAN based image augmentation

The novel proposed CTGAN image augmentation approach is examined in this section. Figure 3 illustrates the workflow architecture of image augmentations. In this research on image augmentations, we have developed an innovative approach, the conditional tabular generative adversarial network (CTGAN) (Lee, Yoon & Kwon, 2020). In CTGAN, the generator neural network focuses on producing realistic synthetic data by learning the data distribution of the input dataset, while the discriminator network evaluates the authenticity of both real and generated data. For image generation, the generator creates images from random noise vectors conditioned on specific input features, and the discriminator classifies images as real or fake based on learned features. Through adversarial training, where the generator enhances its ability to create realistic track images, and the discriminator enhances its capability to distinguish fakes, CTGAN can generate high-quality synthetic images that mimic the characteristics of the original dataset.

Figure 3 The architecture flow of proposed CTGAN-based image argumentation.

The unique combination of our proposed research approach is illustrated in Fig. 4. This architecture provides a comprehensive overview of the fault detection methodology. Initially, the image dataset is augmented using the CTGAN model, and the output of the CTGAN model is subsequently integrated with the Random Forest (RF) model to enhance the classification of railway track fault detection. We generated 2,000 images for each category using the CTGAN approach, balancing the dataset as shown in Fig. 5.

Figure 4 The unique introduced structural details architecture analysis.

Figure 5 Bar chart-based target class analysis.

The CTGAN model consists of two primary components:

Generator (G): The generator maps a latent space Z∼PZ(z) to the image space, conditioned on specific input features: (1) G:Z×C→X′

where: Z∼N(0,I) is a multivariate Gaussian distribution in the latent space.

C represents the conditioning variable (i.e., track fault categories).

X′ is the synthetic railway track image generated by the model.

Discriminator (D): The discriminator distinguishes between real X and synthetic X′ images while preserving the feature distribution:

(2) D:X×C→[0,1]

where D(X,C) represents the probability that X is a real image given condition C.

Objective function: The adversarial loss function follows the standard GAN formulation:

(3) minGmaxDEX∼Pdata(X)[log⁡D(X|C)]+EZ∼PZ(Z)[log⁡(1−D(G(Z|C)))]

where Pdata(X) is the real data distribution, and PZ(Z) is the prior noise distribution.

Once the railway track images are generated using CTGAN, they are integrated into the dataset to enhance railway track fault detection using the RF classifier.

Feature extraction: Each railway track image (either real or synthetic) is transformed into a feature vector F through a feature extraction function Φ:

(4) F=Φ(X)

where Φ:X→Rd extracts a d-dimensional feature vector from the image X.

The extracted feature vectors are passed to the RF classifier, which consists of an ensemble of N decision trees Ti:

(5) Y^=1N∑i=1NTi(F)

where Y^ represents the predicted class label.

Description of artificial intelligence models used

Artificial intelligence-based railway track image classification leverages advanced machine learning algorithms (Younas et al., 2024; Khalid et al., 2024; Rustam et al., 2024; Raza et al., 2023, 2024) to automate the detection and classification of railway track defects from images. By utilizing deep learning techniques, this technology enhances the accuracy and efficiency of maintenance operations, ensuring safer and more reliable railway systems through real-time monitoring and early fault detection. K-neighbors classifier (KNC): is a type of supervised machine learning algorithm, which is a category of non-parametric models (Al-Otaibi et al., 2024). KNC works on a straightforward concept: it determines the classification of a new data point by identifying the majority class among its k closest neighbors. The key parameter, k, determines the number of neighbors to consider. The distance metric, often Euclidean, is used to compute the closeness among points. Given a set of training data (xi,yi) where xi∈Rn expressed the features of the ith sample and yi its associated class label, the classification of a new sample x is determined by:

(6) y=mode({yi:d(x,xi)≤d(x,xj)foratleastkvaluesofj})

where d(x,xi) value is the distance between the sample point x and a point xi in the training set.

Logistic regression (LR): is a statistical method that uses the logistic function to predict the probability of occurrence of an event by fitting image data (Rahmatinejad et al., 2024). It is particularly useful in the situation of fault detection in railway tracks where the outcomes are typically binary as ‘Fault’ or ‘No Fault’. In LR, the probability that an outcome y value is 1 given predictors x is modeled as the logistic function of a linear combination of the predictors. The LR model is given by the following equation:

(7) p(y=1|x)=11+e−(β0+β1x1+⋯+βnxn)

where: – p(y=1|x) data is the probability of the track having a fault given predictors x.

– β0,β1,…,βn are the parameters of the model.

– x1,…,xn are the feature values of the track data.

– e is the base data value of the natural logarithm.

Support vector classification (SVC): is a supervised machine learning method (Shoghi et al., 2024) that can be applied to classify railway track segments as either ‘faulty’ or ‘non-faulty’ based on various input features derived from sensor data, such as vibration levels, acoustic signals, and other track condition indicators. The basic principle behind SVC is to find the ideal hyperplane that best separates the data value points into different classes. For railway track fault detection, the input features of each track segment (data points) are mapped into a high-dimensional data space. SVC then attempts to find the hyperplane that maximally separates the faulty and non-faulty classes. The aim of SVC is to solve the following optimization problem:

(8) minw,b12w⊤w

(9) s.t.yi(w⊤xi+b)≥1,∀i,

where w value is the normal vector to the hyperplane space, b is the bias value term, xi are the feature vectors, yi are the labels associated with each feature vector (1 for non-faulty and −1 for faulty), and i indexes over all the data points.

Random forest classifier (RFC): combines multiple DTs during construction to improve the overall performance of the model (Amiri et al., 2024). It is particularly effective in dealing with high-dimensional datasets and complex data structures, making it suitable for applications like fault detection in railway tracks. RFC functions by creating numerous DTs during the training phase and producing the class that represents the most frequent class predicted by the individual trees. The classification decision in a RFC is typically given by taking the majority vote across all trees. Mathematically, this can be represented as:

(10) y=mode({y1,y2,…,yn})

where y is the predicted class, and y1,y2,…,yn are the classes predicted by each of the n trees in the forest.

Convolutional neural network (CNN): is a deep networks (Lee et al., 2024), most commonly applied to analyzing visual imagery data. They have proven very effective in areas such as image recognition and classification, paving the way for innovative applications in various fields, including railway track fault recognition. In the context of railway track fault detection, CNNs can analyze visual data from track images to identify defects such as cracks, breaks, and other irregularities. The output layer in a CNN typically involves a softmax function. The softmax function converts the output scores from the final layer into probability values. The mathematical expression for the softmax data function is:

(11) P(y=k|x)=ezk∑j=1Kezj

Here, P(y=k|x) is the probability that input x belongs to class k, zk represents the input to the softmax data function from the last fully connected layer of the network, and K is the total values of classes.

Hyperparameter optimization

In our study, we optimized the parameters of various applied machine learning methods through a handcraft recursive training and testing process, utilizing a k-fold validation approach to ensure robust and unbiased model performance. Hyperparameter optimization is a critical step in our methodology, allowing us to fine-tune the models for better accuracy and generalization. Table 2 presents the optimal hyperparameters identified for each method. This rigorous hyperparameter tuning process enabled us to enhance the performance of railway track fault detection. We selected the hyperparameters based on their effectiveness in achieving high performance scores for railway track fault detection.

Table 2 Optimal hyper-parameters tuning analysis.

Methods	Hyperparameters	
RFC	max_depth=50, random_state=0, n_estimators=50	
SVC	penalty=‘l2’, loss=‘squared_hinge’, tol=0.0001, C=1.0, max_iter=1,000	
KNC	n_neighbors=3, leaf_size=30, p=2, metric=‘minkowski’	
LR	penalty=‘l2’, tol=0.0001, C=1.0, solver=‘lbfgs’, max_iter=100	
CNN	activation=‘sigmoid’, padding=“same”, loss = ‘binary_crossentropy’, optimizer = ‘adam’	
CTGAN	epochs=5, samples=2,000	

For CTGAN, we used the following training parameters: batch size = 500, generator learning rate = 2e-4, discriminator learning rate = 2e-4, latent dimension = 128, PacGAN number (pac) = 10, and stopping criteria = fixed epoch limit.

Assessment metrics

To evaluate the performance of the machine learning approaches, several metrics were employed, including accuracy, recall, F1 score, and precision score. The detailed mathematical explanation of each metric used is shown in Table 3. Here, the terms TP refer to true positive, FP refers to false positive, TN refers to true negative, and FN refers to false negative.

Table 3 Formulas for classification metrics.

Metric	Formula details	
Accuracy	TP+TNTP+TN+FP+FN	
Precision	TPTP+FP	
Recall (Sensitivity)	TPTP+FN	
F1 score	2×Precision×RecallPrecision+Recall	

Results and discussions

This section is based on the performance results of machine learning techniques for railway track fault recognition. In our study, we employed several machine learning methods to analyze image data collected from various segments of the railway track. The results and findings suggest that machine learning, particularly deep learning models can revolutionize railway maintenance by providing a scalable and reliable solution for track fault detection.

Computing infrastructure

The experimental setup for this study involves implementing machine learning techniques using the Python programming language. All experiments were conducted in an online environment, specifically Google Colab, which provided a GPU backend with NVIDIA T4 and compute capability of 7.5 to facilitate the computational requirements. The Google Colab environment was configured with a 90-GB value of disk space and a 13-GB value of RAM, ensuring sufficient resources for data processing and model training.

Results of classical CNN

In Fig. 6, the performance of a classical CNN model is evaluated for railway track fault recognition using time series data. The results include metrics such as accuracy, loss, recall, and precision scores, which were monitored throughout the training process. The time series graph revealed that during the initial five epochs, the loss scores for both training and validation were relatively high, indicating initial model instability. Similarly, accuracy, precision, and recall scores were low during these initial epochs. However, as the training progressed, the CNN model updated its weights effectively, leading to a gradual decrease in loss scores and a corresponding improvement in accuracy, precision, and recall. This trend demonstrates the model’s learning curve and its ability to enhance performance over time, ultimately achieving better results in detecting faults in railway tracks.

Figure 6 A time series analysis utilizing a classical CNN model.

The unseen testing results of the classical CNN method, as depicted in Table 4, demonstrate robust performance metrics across different target classes. The overall accuracy achieved is 0.97. When broken down by target class, the CNN method shows a precision score of 0.98, a recall score of 0.97, and an F1 score of 0.97 for the Non-Defective class. For the Defective class, the precision score is 0.96, the recall score is 0.97, and the F1 score is 0.97. The average performance across both classes also yields a precision, recall, and F1 score of 0.97. These results indicate that the classical CNN method is highly effective in accurately distinguishing between defective and non-defective items. The consistent performance across all metrics suggests a balanced model that does not favour one class over the other. However, there is potential for further improvement to push the accuracy beyond the current 0.97.

Table 4 Testing results of classical CNN method.

Accuracy	Target class	Precision	Recall	F1	
0.97	Non-Defective	0.98	0.97	0.97	
	Defective	0.96	0.97	0.97	
	Average	0.97	0.97	0.97	

Results of machine learning methods

Table 5 presents a comparative analysis of different machine learning methods used for detecting faults in railway tracks. The KNC attained an overall accuracy score of 96%, with precision, recall, and F1 scores each at 0.96 for both defective and non-defective track conditions, reflecting strong balanced performance across categories. The LR and SVC methods both significantly outperformed KNC, achieving an overall accuracy of 99% along with perfect precision, recall, and F1 scores of 0.99 for both defective and non-defective states, demonstrating high efficacy in fault detection. However, the standout performer is the RFC, which not only matched the highest accuracy of 99% but achieved a perfect precision and recall score of 1.00 for defective track conditions. This indicates that the proposed RFC can identify all actual defective cases without falsely categorizing any non-defective tracks as defective, which is critical in preventing both unnecessary maintenance costs and potential accidents due to missed defects. For non-defective tracks, RFC recorded precision and recall scores of 0.99, maintaining exceptional reliability in identifying true negative cases. These results highlight the effectiveness of RFC in railway track fault detection, suggesting it is the most robust model among those tested.

Table 5 Results of applied machine learning techniques.

Method	Accuracy	Target	Precision	Recall	F1	
KNC	0.96	Defective	0.98	0.94	0.96	
		Non-Defective	0.94	0.98	0.96	
		Average	0.96	0.96	0.96	
LR	0.99	Defective	0.99	0.99	0.99	
		Non-Defective	0.99	0.99	0.99	
		Average	0.99	0.99	0.99	
SVC	0.99	Defective	0.99	0.99	0.99	
		Non-Defective	0.99	0.99	0.99	
		Average	0.99	0.99	0.99	
RFC	0.99	Defective	1.00	1.00	1.00	
		Non-Defective	0.99	0.99	0.99	
		Average	0.99	0.99	0.99	

Figure 7 presents the confusion matrix validation analysis of the applied approaches for railway track fault detection. The analysis highlights the error rate of each method. The KNC had 31 incorrect predictions, LR had 8, RF had 4, and SVC had 9 incorrect predictions. This analysis validates that the proposed RF model achieved a minimal error rate, indicating its superior performance in railway track fault classification with a high correct classification rate. The significantly lower error rate of the RF model underscores its effectiveness and reliability in identifying faults accurately compared to the other methods evaluated.

Figure 7 The confusion matrix validation analysis.

K-fold cross validations analysis

For the validation of machine learning methods performance for railway track fault detection, a K-fold validation technique is employed. The analysis in Table 6 is conducted using a 10-fold validation scheme for each method to ensure the stability and generalizability of the results. The KNC achieved an accuracy of 96% with a standard deviation of ±0.0092, indicating high performance with relatively low variability in prediction outcomes across different subsets. Logistic regression (LR) and SVC both exhibited an even higher accuracy of 98%, with standard deviations of ±0.0063 and ±0.0073 respectively, suggesting robustness in model predictions. Notably, RFC demonstrated the highest accuracy among the evaluated models, achieving a score of 99% with a standard deviation of ±0.0081. This result underscores the effectiveness of RFC in handling complex patterns in data about railway track anomalies, thus providing a highly reliable tool for fault detection in critical transportation infrastructure.

Table 6 K-fold validations results analysis.

Method	K-fold	Accuracy	Standard deviations (±)	
KNC	10	0.96	0.0092	
LR	10	0.98	0.0063	
SVC	10	0.98	0.0073	
RFC	10	0.99	0.0081	

Computations complexity analysis

Table 7 provides a summary of the computational complexity analysis for the methods applied. The runtime computations, measured in seconds, reveal significant differences in the performance of each method. The KNC demonstrated the fastest computation time at 0.02110 s, however with a low accuracy score. The LR method required 0.28747 s, showing a moderate computational demand. The RFC, while offering robust predictive power, exhibited the highest computational cost at 2.70125 s. These results highlight the trade-offs between computational efficiency and potential predictive accuracy of railway track fault detection.

Table 7 The computation analysis of applied methods.

Method	Computations runtime (s)	
KNC	0.02110	
LR	0.28747	
RFC	2.70125	
SVC	0.04133	

State of the art comparison

The state-of-the-art results comparison of our proposed novel study is shown in Table 8. Our research demonstrates superior performance when compared to existing methodologies. Shafique et al. (2021) achieved a 97% accuracy using random forest (RF) and decision tree (DT) models. Eunus et al. (2024) reported a 93% accuracy with their ECARRNet approach. Mahale & Gaikwad (2024) obtained a 98% accuracy utilizing a combination of InceptionV3, ResNet50V2, and VGG16 models. In contrast, our proposed model, which integrates CTGAN with RF, achieved a remarkable 99% accuracy for railway track fault detection.

Table 8 State-of-the-art studies performance comparisons.

References	Model	Reported accuracy	
Shafique et al. (2021)	RF and DT	97%	
Eunus et al. (2024)	ECARRNet	93%	
Mahale & Gaikwad (2024)	InceptionV3, ResNet50V2, and VGG16	98%	
Proposed	CTGAN+RF	99%	

In addition, a recent study (Bird et al., 2022) used GAN data augmentation for fruit defect image classification, achieving a highest accuracy of 88%. This performance is lower compared to our proposed approach.

Ablation study

In this study, we conducted an ablation study to evaluate the performance of classical CNN methods and a proposed RFC approach in detecting defective track conditions. The classical CNN method achieved an overall accuracy of 0.97. When examining the performance by target class, the CNN method demonstrated a precision score of 0.98, a recall score of 0.97, and an F1 score of 0.97 for the Non-Defective class. For the Defective class, the precision score was 0.96, the recall score was 0.97, and the F1 score was 0.97. Despite these high scores, there is a noticeable performance gap using the classical approach, particularly in achieving perfect classification.

In contrast, the proposed approach utilizing RFC emerged as the standout performer, as illustrated in Fig. 8. It not only matched the highest accuracy of 99% but also achieved perfect precision and recall scores of 1.00 for defective track conditions. This indicates that the proposed RFC approach can identify all actual defective cases without falsely categorizing any non-defective tracks as defective, thus demonstrating its superior capability in precise defect detection and reducing false positives. The results underscore the efficacy of the RFC approach in improving detection accuracy and reliability compared to the classical CNN method.

Figure 8 The ablation study results analysis.

Statistical significance analysis

Our research evaluates the effectiveness of the proposed CTGAN+RF approach using statistical analysis. Performance differences with other models are assessed via t-statistics and p-values. CTGAN+RF significantly outperforms CNN (t = −18.989, p = 6.0046) and KNC (t = −25.999, p = 7.2749), confirming its superiority.

Limitations

In our proposed model, the KNC method demonstrated the fastest computation time of 0.02110 s but with a notably low accuracy score. This suggests that while KNC is efficient in terms of speed, it may not be the most reliable option for railway track fault detection. The LR method required 0.28747 s, indicating a moderate computational demand while balancing efficiency with accuracy. On the other hand, the RFC offered robust predictive power but at the cost of the highest computational time of 2.70125 s.

These findings underscore the inherent trade-offs between computational efficiency and predictive accuracy in detecting railway track faults. Recognizing these limitations, future work will focus on optimizing the model’s layer stack to simplify the architecture, thereby reducing computational costs while striving to maintain or improve accuracy.

Implementation challenges

Implementing the proposed model in a real-time railway monitoring system is feasible, provided that sufficient computational resources and real-time data acquisition mechanisms are available. The model can be deployed as part of an edge computing system installed on railway inspection vehicles or integrated into cloud-based platforms for centralized monitoring and decision-making. The following challenges can be faced: The effectiveness of the model relies on high-quality, real-time data acquisition from sensors, cameras, and drones, which may be affected by environmental conditions, hardware limitations, and data transmission delays.

Railway infrastructure conditions evolve due to wear, weather effects, and repairs.

Running complex machine learning models on real-time railway monitoring systems requires computational efficiency, particularly if resources are limited in mobile setups.

Conclusions

This research developed an advanced generative neural network for efficient railway track fault detection. We proposed a novel CTGAN-based image augmentation approach to producing realistic synthetic image data using railway track images. We developed five advanced neural network techniques for comparison with railway track image classification. The implemented RF method stood out state-of-the-art study with the highest accuracy score of 0.99 for railway track fault detection. Hyperparameter optimization is applied to achieve optimal performance, and the performance is evaluated using the k-fold validation approach.

In future research, a promising avenue involves the development of a camera-based framework designed to detect faults in railway tracks in real-time. This innovative approach utilizes the proposed image recognition approach integrated into a backend system. The framework aims to enhance railway safety by continuously monitoring track conditions and promptly issuing alerts upon detecting any faults or anomalies.

Supplemental Information

Supplemental Information 1 Coding experiments.

Supplemental Information 2 Readme.

Supplemental Information 3 Rebuttal Authorship change.

Additional Information and Declarations

Competing Interests

The authors declare that they have no competing interests.

Author Contributions

Ali Raza conceived and designed the experiments, performed the experiments, analyzed the data, performed the computation work, prepared figures and/or tables, authored or reviewed drafts of the article, and approved the final draft.

Rukhshanda Sehar conceived and designed the experiments, analyzed the data, performed the computation work, prepared figures and/or tables, authored or reviewed drafts of the article, and approved the final draft.

Abdul Moiz performed the experiments, prepared figures and/or tables, authored or reviewed drafts of the article, and approved the final draft.

Ala Saleh Alluhaidan conceived and designed the experiments, analyzed the data, prepared figures and/or tables, authored or reviewed drafts of the article, and approved the final draft.

Sahar A. El-Rahman performed the experiments, prepared figures and/or tables, authored or reviewed drafts of the article, and approved the final draft.

Diaa Salama AbdElminaam conceived and designed the experiments, analyzed the data, prepared figures and/or tables, authored or reviewed drafts of the article, and approved the final draft.

Data Availability

The following information was supplied regarding data availability:

The dataset is available at Kaggle: Adnan A, Salman Ibne Eunus SH. 2024. Railway track fault detection—Kaggle. version 2. https://www.kaggle.com/datasets/gpiosenka/railway-track-fault-detection-resized-224-x-224.

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
