# Peer review of "Novel conditional tabular generative adversarial network based image augmentation for railway track fault detection"

_PeerJ Computer Science, doi:10.7717/peerj-cs.2898_

## Round 0.1 · original submission · Major Revisions

Please revise the paper according to the reviewer's comments

Reviewer 1 ·

Basic reporting

The paper generally demonstrates good scientific reporting, but there are several areas that require improvement. The introduction provides adequate context about railway track fault detection and its importance, but would benefit from a clearer statement of the specific research gap being addressed. While the literature review covers relevant work, it could be better organized thematically rather than chronologically to help readers understand the evolution of approaches in this field. The motivation for using CTGAN-based image augmentation is not sufficiently explained in the introduction - the authors should elaborate on why this specific approach was chosen over other potential solutions.

The formal methodology section needs more rigorous mathematical definitions, particularly regarding the CTGAN architecture and its integration with the Random Forest classifier. While equations are provided for the machine learning methods used, the novel aspects of the CTGAN implementation lack detailed mathematical formulation. The experimental setup description in Section 4.1 is quite brief and should include more details about the dataset preprocessing steps and implementation specifics to ensure reproducibility.

The paper's structure generally follows PeerJ standards, though the Results and Discussion sections could be better integrated. Currently, some results are presented without sufficient discussion of their implications, while other findings are discussed without clear reference to the supporting data. The English used throughout is generally professional and clear, though there are occasional awkward phrasings that should be revised for clarity (for example, in the abstract: "To address these challenges, efficient techniques, such as artificial intelligence-based image classification, are being explored" could be more precisely worded). Additionally, some technical terms are used without proper introduction or definition, which could make the paper less accessible to readers from adjacent fields.

Experimental design

The experimental design of this study falls within the scope of PeerJ Computer Science, but several methodological aspects require clarification and enhancement. While the authors present a novel approach combining CTGAN with Random Forest classification, the implementation details of the CTGAN architecture are insufficiently described. The paper mentions that 2,000 images were generated for each category using CTGAN, but lacks crucial details about the network architecture, training parameters, and stopping criteria that would be necessary for reproduction. Furthermore, while the README provides some code context, the hyperparameter optimization process described in Section 3.5 needs more detailed explanation of the search space and selection criteria used.

The data preprocessing section is particularly thin, with minimal discussion of important aspects such as image normalization techniques, handling of class imbalances, and validation of the synthetic data quality. The authors should elaborate on how they ensured the generated synthetic images maintained the essential characteristics of real railway track defects. The computing infrastructure is briefly mentioned (Google Colab with 90GB disk space and 13GB RAM), but information about the GPU specifications and runtime environment configurations is missing. While the evaluation metrics are well-defined mathematically in Table 3, the justification for choosing these specific metrics and their relevance to the railway track fault detection context could be strengthened.

The citation practices are generally appropriate, though some recent developments in CTGAN applications for image generation could be better acknowledged. The comparative analysis with state-of-the-art methods is present but could be more comprehensive, particularly in discussing the practical implications of the achieved performance improvements. The authors should also consider adding a discussion about the limitations of their approach, especially regarding computational requirements and potential deployment challenges in real-world railway systems. Additionally, while the code is mentioned to be available, the paper would benefit from including a permanent repository link for long-term accessibility of the implementation.

Validity of the findings

The paper presents potentially significant findings in railway track fault detection, but several aspects of result validation require attention. The authors claim superior performance of their CTGAN-RF hybrid approach, achieving 99% accuracy, but the statistical significance of this improvement over existing methods is not rigorously demonstrated. While the ablation study provides some insights into the contribution of different components, it doesn't fully explore the impact of the CTGAN-based augmentation on the final model performance. The comparison with state-of-the-art methods shows improved accuracy but lacks discussion of practical considerations such as inference time and resource requirements that would be crucial for real-world deployment.

The experimental evaluation is generally thorough, with appropriate use of k-fold cross-validation and multiple performance metrics. However, the robustness of the model under different operational conditions (varying lighting, weather, or track conditions) is not adequately addressed, which is crucial for practical railway applications. The authors meet most of their stated goals from the introduction, particularly in demonstrating the effectiveness of their hybrid approach, but the broader impact on railway maintenance practices could be better articulated. The computational complexity analysis provides valuable insights, but the trade-off between accuracy and processing time needs more thorough discussion, especially considering the real-time requirements of railway track inspection systems.

While the conclusions are generally well-supported by the results, the limitations section could be more comprehensive. The authors briefly mention computational constraints but should expand on other potential limitations such as the generalizability of synthetic data, the impact of different track types or environmental conditions, and challenges in deploying such a system in practice. Future research directions are suggested regarding a camera-based framework but could be more specific about addressing current limitations and expanding the application scope. Additionally, the novelty of the CTGAN-based approach is established, but its specific advantages over traditional augmentation techniques could be more clearly demonstrated through comparative analysis.

Reviewer 2 ·

Basic reporting

Railway track fault recognition is a critical aspect of railway maintenance, aiming to identify and rectify
defects such as cracks, misalignments, and wear on tracks to ensure safe and efficient train operations. This paper proposes a novel conditional tabular generative adversarial network (CTGAN)-based image augmentation approach to producing realistic synthetic image data using railway track images. The main pipeline of this paper is easy to follow. There are some issues needed to be clarified:
1. What are the current problems of road defect detection that the author should give in the introduction section? How do the authors solve this problem?
2. The introduction and summary of relevant work should be more standardized. The current version is too rudimentary.
3. The method section of this article lacks the necessary formulas and the necessary descriptions.

Experimental design

The experimental part lacks the necessary comparison with other advanced methods.

Validity of the findings

no comment.

Additional comments

no comment.

·

Basic reporting

This paper proposes an innovative and well-structured approach to railway track fault detection. It focuses on developing advanced artificial intelligence techniques, including a Conditional Tabular Generative Adversarial Network (CTGAN), to enhance railway track fault detection through realistic and synthetic image augmentation.

Experimental design

- The CTGAN approach is presented as a key contribution for generating synthetic data. Could you elaborate on the effectiveness of CTGAN in generating realistic track images? How did you ensure that the synthetic images accurately represent real-world track defects?

Validity of the findings

Considering the results, how feasible is it to implement the proposed models in a real-time railway monitoring system? What are the challenges in integrating these machine learning models into a live railway system, and how would you address them?

You mention that the KNC method is fast but has low accuracy. How do you plan to mitigate the impact of KNC's low accuracy on the reliability of railway track fault detection? Could the speed of KNC be combined with another method to improve its overall performance?

Additional comments

- The figures must be in the order of appearance
- Figure 1 is not referenced in the text.
- Verify the size of the text.

Reviewer 4 ·

Basic reporting

(1) The manuscript lacks a clear explanation of the practical engineering implications of the proposed research. Despite claims of improving operational efficiency, reducing maintenance costs, and enhancing the safety and reliability of rail transportation, the engineering relevance remains ambiguous.
(2) The novelty of using CTGAN for image augmentation is overstated. GAN-based augmentation is a well-documented approach, and the authors fail to clearly establish how their CTGAN implementation is distinct from prior works.
(3) The literature review lacks depth and fails to clearly position the proposed research within the context of existing work. Key advancements in railway fault detection are either briefly mentioned or overlooked.
(4) The listed contributions (e.g., "proposing CTGAN-based augmentation") are broad and do not specify actionable advancements or challenges overcome in railway fault detection.

Experimental design

(1) The description of the image preprocessing steps is overly vague. Phrases like "basic image processing" need to be elaborated with specific techniques, their parameters, and the rationale for their use.
(2) The computational cost of using CTGAN and Random Forest is highlighted but inadequately compared with alternative methods. This omission limits the practical applicability of the proposed approach. Hyperparameter tuning is described but lacks sufficient methodological detail. No reasoning for chosen parameters is provided.

Validity of the findings

(1) Details about the dataset are insufficient. Aside from referencing a Kaggle dataset, critical information such as the number of original images, diversity, and specific sources is missing. This lack of transparency raises concerns about the generalizability and robustness of the findings.
(2) The manuscript does not adequately discuss how CTGAN-generated images avoid replicating biases in the original dataset, which could affect model performance.
(3) Comparisons to state-of-the-art methods are insufficient. There is little to no discussion of statistical significance, and no baseline non-GAN augmentation methods are included for contrast.

Additional comments

This manuscript investigates the use of CTGAN for synthetic image data generation to improve railway track fault detection using machine learning models. While the research aligns with the scope of PeerJ Computer Science in exploring AI applications, the manuscript requires substantial revision to meet the publication standards of a high-impact journal. The above issues should be addressed to enhance the manuscript's rigor and contribution to the field of AI in railway systems.

---

## Round 0.2 · accepted · Accept

According to the comments of reviewers and your reply, after comprehensive consideration, it is decided that accept.

Reviewer 2 ·

Basic reporting

The authors have addressed the reviewer's comments and I have no further comment.

Experimental design

The authors have addressed the reviewer's comments and I have no further comment.

Validity of the findings

The authors have addressed the reviewer's comments and I have no further comment.

Reviewer 4 ·

Basic reporting

no comment'

Experimental design

no comment'

Validity of the findings

no comment'

Additional comments

The authors have made satisfactory revisions in response to this reviewer’s previous comments, significantly enhancing both the clarity and quality of the manuscript. After addressing the following remaining minor issue, this reviewer recommends accepting the manuscript for publication.
The revised manuscript contains an excessive number of subsections, which disrupts the logical flow and readability of the text. For instance, subsection 2.1, subsection 3.1-3.6 and subsection 4.1-4.10.